# Conductive carbon nanofiber interpenetrated graphene architecture for ultra-stable sodium ion battery

Mingkai Liu[1,7], Peng Zhang[1,7], Zehua Qu[2], Yan Yan[1], Chao Lai [1], Tianxi Liu[3,4,5] & Shanqing Zhang[6]

Long-term stability and high-rate capability have been the major challenges of sodium-ion batteries. Layered electroactive materials with mechanically robust, chemically stable, electrically and ironically conductive networks can effectively address these issues. Herein we have successfully directed carbon nanofibers to vertically penetrate through graphene sheets, constructing robust carbon nanofiber interpenetrated graphene architecture. Molybdenum disulfide nanoflakes are then grown in situ alongside the entire framework, yielding molybdenum disulfide@carbon nanofiber interpenetrated graphene structure. In such a design, carbon nanofibers prevent the restacking of graphene sheets and provide ample space between graphene sheets, enabling a strong structure that maintains exceptional mechanical integrity and excellent electrical conductivity. The as-prepared sodium ion battery delivers outstanding electrochemical performance and ultrahigh stability, achieving a remarkable specific capacity of 598 mAh g$^{-1}$, long-term cycling stability up to 1000 cycles, and an excellent rate performance even at a high current density up to 10 A g$^{-1}$.

[1] School of Chemistry & Materials Science, Jiangsu Key Laboratory of Green Synthetic Chemistry for Functional Materials, Jiangsu Normal University, 221116 Xuzhou, P. R. China. [2] State Key Laboratory of Molecular Engineering of Polymers, Department of Macromolecular Science, Fudan University, 200433 Shanghai, P. R. China. [3] Key Laboratory of Synthetic and Biological Colloids, Ministry of Education, School of Chemical and Material Engineering, Jiangnan University, 214122 Wuxi, P. R. China. [4] State Key Laboratory for Modification of Chemical Fibers and Polymer Materials, College of Materials Science and Engineering, Donghua University, 201620 Shanghai, P. R. China. [5] Key Laboratory of Materials Processing and Mold (Zhengzhou University), Ministry of Education, 450002 Zhengzhou, P. R. China. [6] Centre for Clean Environment and Energy, Griffith School of Environment, Griffith University, Gold Coast, QLD 4222, Australia. [7] These authors contributed equally: Mingkai Liuand, Peng Zhang. Correspondence and requests for materials should be addressed to C.L. (email: laichao@jsnu.edu.cn) or to T.L. (email: txliu@jiangnan.edu.cn) or to S.Z. (email: s.zhang@griffith.edu.au)

odium ion batteries (SIBs), as one of the most promising candidates among next-generation energy storage systems, have attracted tremendous interest due to sodium's natural abundance and ready accessibility[1–11]. However, compared to lithium ions (0.59 Å), the larger diameter (0.99 Å) of sodium ions (Na+) limits the number of suitable electroactive materials and hinders the electrochemical interfacial reaction kinetics. As such, owing to the sluggish Na+ insertion/extraction efficiency, the poor rate performance of SIBs has been well recognized as an inherent challenge[12–15]. In the last decade, much effort has been devoted to developing promising 2D structural anode materials, such as phosphorus[16], carbonaceous materials[17,18], metallic alloys, and two-dimensional carbides (MXenes)[19,20], to improve the electrochemical performances of SIBs and promote their practical application[21–25]

Among the investigated electrode materials, 2D molybdenum disulfide (MoS2), a layered transition-metal-dichalcogenide (TMD) material with S–Mo–S motifs stacked together by Van der Waals forces, is considered one of the most promising anode materials for SIBs[26–28]. MoS2 materials can be further modified as intercalation-type anode materials with expanded d-spacing to improve the electrochemical performances of state-of-art anodes. However, MoS2-based electrodes exhibit poor rate capability and fast capacity fading upon cycling due to low electrical conductivity and the huge volume variations during charge/discharge process[29–31]. Incorporation of MoS2 nanomaterials into highly conductive carbonaceous matrices was suggested as an effective way to address this problem[32–35]. To date, several MoS2-carbon hybrid materials have been developed, such as MoS2-graphene composites, MoS2-CNT hybrids, and MoS2-carbon spheres[36–38]. The electrochemical performance, in terms of specific capacity, has been significantly improved due to the excellent electrical conductivity offered by the carbon matrices ensuring rapid electron transfer in the charge/discharge processes. However, there is still much room for improvement in terms of rate capability and stability of these anode materials. Thus, development of MoS2/carbon hybrids with resilient porous structure for rapid ionic transport and storage is urgently needed and of great importance.

Graphene is considered a most promising carbon material due to its inherent advantages, including large surface area, high conductivity and exceptional mechanical strength[39–42]. However, such advantages would vanish if the graphene sheets restack. Carbon nanotubes (CNTs) and carbon nanofibers (CNFs) are used to prevent the restacking of graphene sheets but the improvement is very limited. Such simple hybrids offer limited surface area enhancement and limited channels for ionic transfer due to the fact that the CNTs and CNFs are in parallel with the graphene plane. It is extremely challenging to steer the CNFs to vertically penetrate through the graphene plane. To the best of our knowledge, this vertical penetration has not been achieved in the literature.

In this work, inspired by the floors-and-pillars concept in construction (Supplementary Fig. 1), we design and develop a robust 3D conductive CNFs interpenetrated graphene (CNFIG) architecture by directing CNFs to penetrate through the graphene sheets. MoS2 nanoflakes are then in situ deposited on the surface of the CNFIG framework, producing a MoS2@CNFIG hybrid. It is envisaged that the MoS2@CNFIG hybrid possess several important advantages due to its unique structural characteristics, including: (i) excellent transportation channels can be integrally preserved during the rapid penetration of electrolyte and rapid transfer of ions for long-term cycles; greatly contributing to the high rate performance of the assembled batteries; (ii) the CNFs can simultaneously act as supporting pillars between different carbon layers and play an important role in rapid transfer of electrons; and (iii) due to their homogeneous deposition, all the active sites of MoS2 nanosheets can be thoroughly exposed to the electrolyte and Na+, which produces high energy density for the MoS2@CNFIG hybrid. Furthermore, the MoS2@CNFIG hybrid in this work could inspire more electrode designs with stable inner structures with high rate performance and long-term cycling stability.

## Results

**Structural characterizations of CNFIG architecture.** The preparation of the hierarchical CNFIG architecture is schematically illustrated in Fig. 1a. CNFs with an average diameter of 1 μm (Fig. 1b), were prepared from the PAA fiber membranes (Supplementary Fig. 2). The carbon fiber networks were derived from the electrospun PAA fiber networks (Supplementary Fig. 3) under chemical imidization and high-temperature carbonization. Here, the PAA matrix was polymerized by ODA and PMDA monomers (Supplementary Fig. 4). CNFs were dispersed within graphene oxide solution under strong sonication and stirring. Graphene oxide (GO) sheets with large domain size were presented in Fig. 1c. PAA powder (Supplementary Fig. 5) can be redispersed into ultrapure water with the assistance of triethylamine (TEA), forming the PAA chains. Vertically aligned channels can be clearly observed in the overall image of the carbon networks (Fig. 1d). These channels can contribute to the rapid penetration of electrolyte and quick transfer of Na+. More interestingly, numerous CNFs are perpendicularly placed across the aligned channels acting as supporting pillars between the adjacent carbon layers (Fig. 1e). Detailed morphological information can be found in the SEM image at high magnification, (Fig. 1f). Most of the CNFs are inserted through the carbon layers. This CNFIG aerogel with extremely stable channels can provide excellent transfer pathways for electrolyte and ions. Meanwhile, the supporting CNF pillars can further act as conductive bridges to accelerate the transfer of electrons, which will contribute to the electrochemical performance of the electrodes.

The vertically aligned channels, as well as the porous morphology of CNFIG, can be maintained even under large pressure. Figure 1g schematically illustrates the compressible capability of the robust CNFIG aerogel due to the excellent supporting/interconnecting effect of the inserted CNFs. Figure 1h presents images of CNFIG aerogel being compressed and released. The CNFIG aerogel can completely recover to its original shape without any mechanical fracture even after being compressed up to 90%. The compressive stress-strain curves at the set strains (ε) of 60, 70, 80, and 90% for CNFIG aerogel are shown in Supplementary Fig. 6. A linear elastic region at ε < 60% and the densification region at ε > 60% can be detected in the compressive stress-strain curves. A much higher compressive stress of approximate 0.25 MPa can be achieved at the set strain ε = 90%. In addition, the cyclic stress-strain curves of CNFIG at a maximum strain of 90% were cycled more than 100 times. The stable and constant stress-strain curves in the 1st, 30th, and 100th cycles further confirm the recoverability of this CNFIG aerogel. Meanwhile, the CNFIG aerogel that was compressed 100 times shows a stable layered morphology with constant channels and supporting CNF pillars (Supplementary Fig. 7), which further confirms its robust capability[43]. CNFs with larger diameter of 900 nm can also be interpenetrated across the graphene sheets (Supplementary Fig. 8), which illustrates the general application of this fabrication strategy.

**Formation mechanism of CNFIG.** Figure 2 demonstrates the proposed formation mechanism of CNFIG. With prepared GO/PAA/CNF solution vertically dipped into the liquid nitrogen, ice pillars will be homogeneously grown on a vertical direction inside

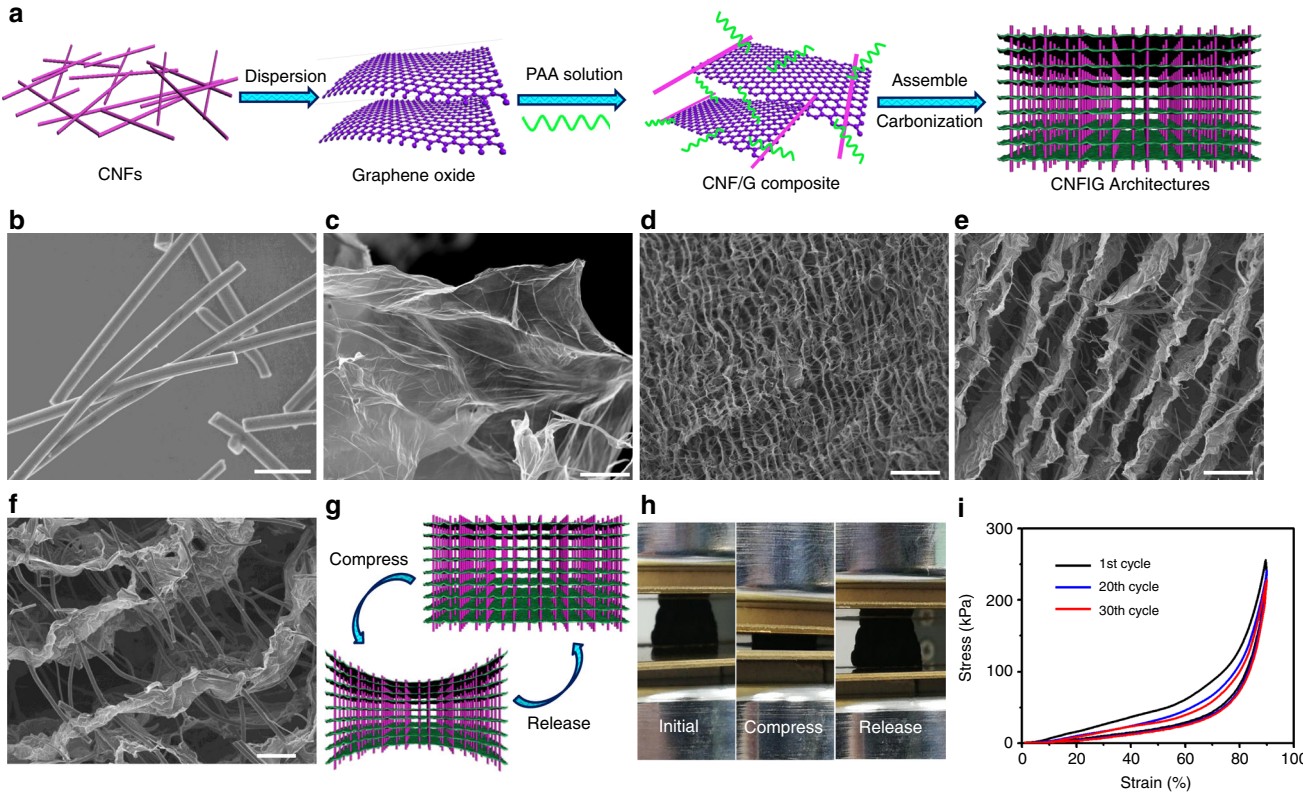

**Fig. 1** Structure design and morphology of CNFIG. **a** Schematic illustration of the synthesis of ultra-stable CNFIG architecture based on CNFs and graphene sheets; SEM images of **b** CNFs (scale bar = 5 μm) and **c** GO sheets (scale bar = 5 μm); SEM images of CNFIG architectures at **d** low (scale bar = 30 μm), **e** medium (scale bar = 20 μm), and **f** high (scale bar = 10 μm) magnifications; **g** schematic representation of the compressible ability of CNFIG; **h** digital photos of the CNFIG architecture being compressed; **i** Stress-strain curves of CNFIG at the 1st, 20th, and 30th cycles

the formed block (Fig. 2a). Aligned pores will be created after the ice pillars were removed, resulting from the freeze-drying treatment. Here, the perpendicular alignment of CNFs might be resulted from several factors. Firstly, CNFs and GO sheets can be tightly connected with the assistance of PAA molecular chains due to the existed functional groups on their surface or on the chains of PAA (such as –COOH, –OH, etc.) (Fig. 2a), apart from the electrostatic interaction between CNFs and GO sheets. With ice pillars growing, PAA/GO matrix will be pushed aside to form the precursor for carbon layers. During this process, the long CNFs that attached on different GO sheets are directed to a direction vertical to the ice pillars due to the pull force (*f*) as illustrated in Fig. 2b. Here, it should be stated that most of the CNFs can be forced to perpendicularly across the carbon layers, however, still leaving a little to be attached on the surface of the formed PAA/GO mixture layers, as seen in Fig. 2c. Secondly, the existence of PAA molecular chains and GO sheets plays an important role in the formation of CNFIG, as demonstrated in Fig. 2d–g. Disordered pores with large size will be formed if only PAA polymer matrix was used (Fig. 2d). And this phenomenon can be ascribed to the typical self-assembled aerogel of polymers as a result of the freeze treatment in liquid nitrogen[44,45]. However, with the assistance of introduced raw GO sheets, clearly aligned carbon layers or pores can be achieved in the obtained carbon aerogel due to the interfacial interaction of PAA chains and GO sheets (Fig. 2e) in the preparation process. The important role of PAA chains can also be demonstrated, as seen in Fig. 2f, carbonic foam with disordered porous structures will be resulted if only pristine GO and CNFs were used. This result further illustrates that the crosslink of PAA polymer chains on the surface of CNFs and GO sheets has a vital function to generate the

aligned pores and carbon layers. Also, this result can be an effective confirmation of the interfacial interaction between CNFs and GO sheets. Meanwhile, if GO sheets was not used, CNF@PAA mixture that used as precursor will only create a similar self-assembled morphology as that of pure PAA matrix, as seen in Fig. 2g. Thirdly, the length of CNFs can also determine the final morphology of CNFIG. Here, the average length of the used CNFs is about 30–40 μm. If shorter CNFs were used (i.e. 2–3 μm), the obtained CNFIG architecture (Fig. 2h) will only has a similar morphology as that of GO@PAA (Fig. 2e). The shorter CNFs will only be attached on the surface of carbon layers yet not perpendicularly crossed them, as demonstrated in Supplementary Fig. 9. These results demonstrate the formation mechanism of CNFIG that PAA molecular chains, GO sheets, the length of CNFs and ice pillars can play a crucial role in determining the final morphology of CNFIG architectures.

**Fabrication and structural characterization of MoS₂@CNFIG hybrid**. The CNFIG aerogel also possesses a high electrical conductivity, up to 15.6 S cm⁻¹, as confirmed by the electrical current-voltage curve (Supplementary Fig. 10). Meanwhile, a break copper wire can be connected by a piece of CNFIG in a turn-on electrical circle (Supplementary Fig. 11), demonstrating the good electrical conductivity of this CNFIG matrix. Here, the developed CNFIG aerogel is used as a conductive template for the homogeneous deposition of layered MoS₂ nanoflakes (Fig. 3a). The robust MoS₂@CNFIG hybrid is fabricated via a versatile interfacial deposition approach and subsequent high-temperature treatment. Due to the homogeneous deposition reaction, layered MoS₂ nanoflakes can be uniformly anchored on the carbon layers

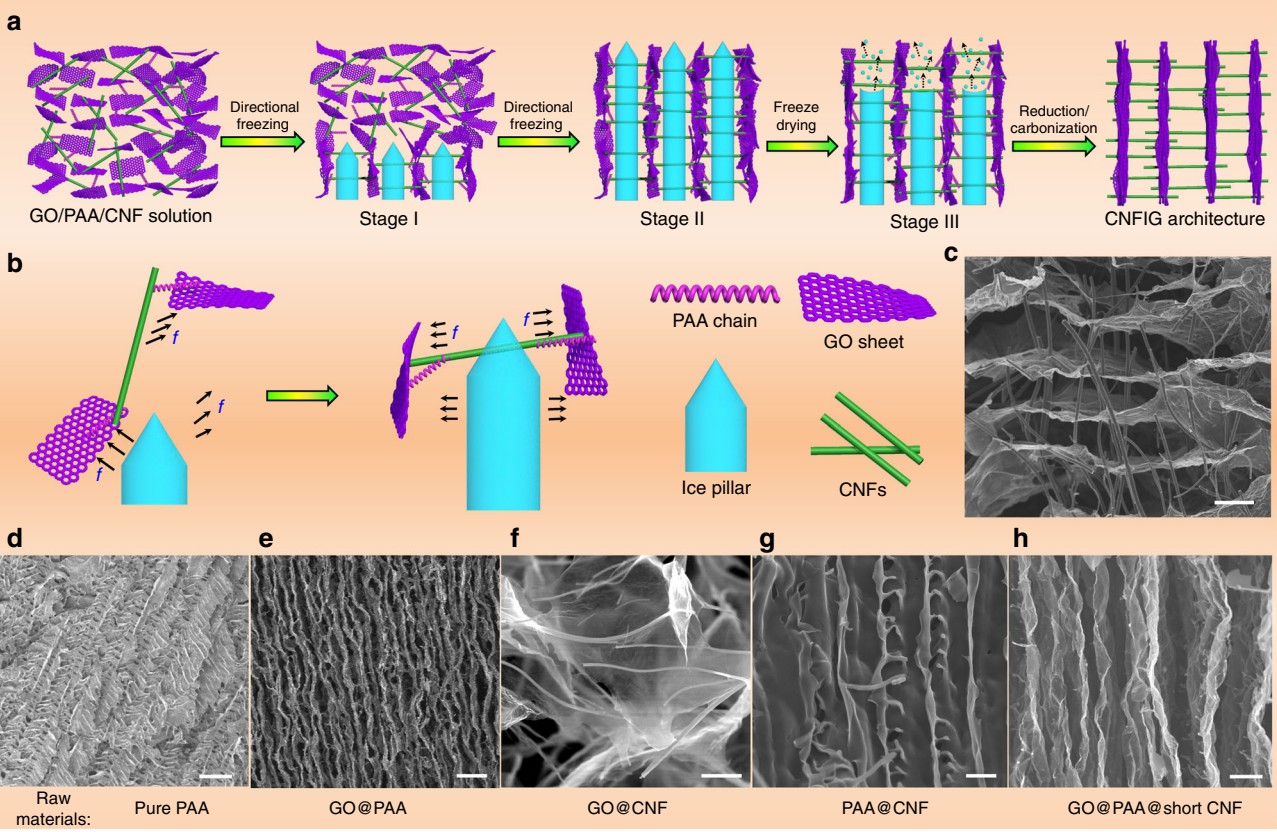

**Fig. 2** The proposed formation mechanism of CNFIG: **a** the directional freezing leads to the creation of ice pillars, the parallel alignment of graphene sheets, and the aligned channels for CNFs. **b** the mechanism of the perpendicular alignment of the CNFs by the parallel alignment of graphene sheets. SEM images of the CNFIG: **c** CNFIG and carbon aerogels prepared from raw materials of **d** pure PAA matrix, **e** GO@PAA, **f** GO@CNF, **g** PAA@CNF, **h** GO@PAA@short CNF. (Scale bars: **c** 10 μm, **d** 30 μm, **e** 10 μm, **f** 4 μm, **g**. 5 μm, **h** 5 μm)

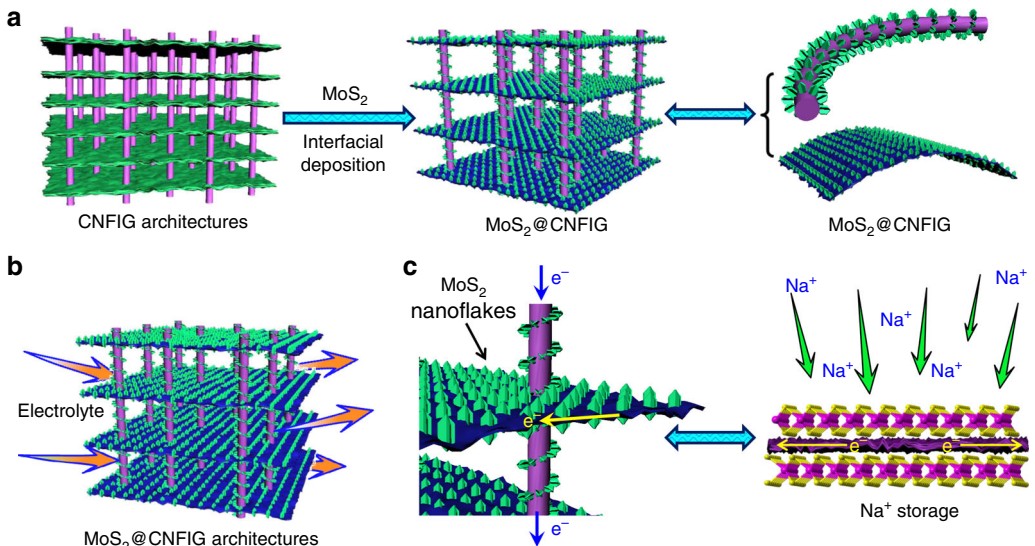

**Fig. 3** Schematic illustration of the preparation of MoS₂@CNFIG hybrid and sodium ions storage. **a** Schematic illustration of the homogeneous deposition of layered MoS₂ nanoflakes on CNFIG matrix, **b** porous morphology of MoS₂@CNFIG hybrid accelerates the rapid penetration of electrolyte, and **c** quick sodiation/desodiation and large sodium ions storage capabilities of MoS₂@CNFIG hybrid due to excellent electrical conductivity

and the CNF bridges. Here, the vertically aligned channels in the MoS₂@CNFIG hybrid ensure rapid penetration of the electrolyte and also contribute to the rapid transfer of sodium ions (Fig. 3b). The excellent electrical conductivity of CNFIG matrix can also provide effective pathways for fast electron transportation (Fig. 3c), which will benefit the insertion/extraction of sodium ions.

The vertically aligned channels created inside the 3D carbonic CNFIG networks are integrally maintained in the developed MoS₂@CNFIG nanohybrid (Fig. 4a). SEM images focusing on the

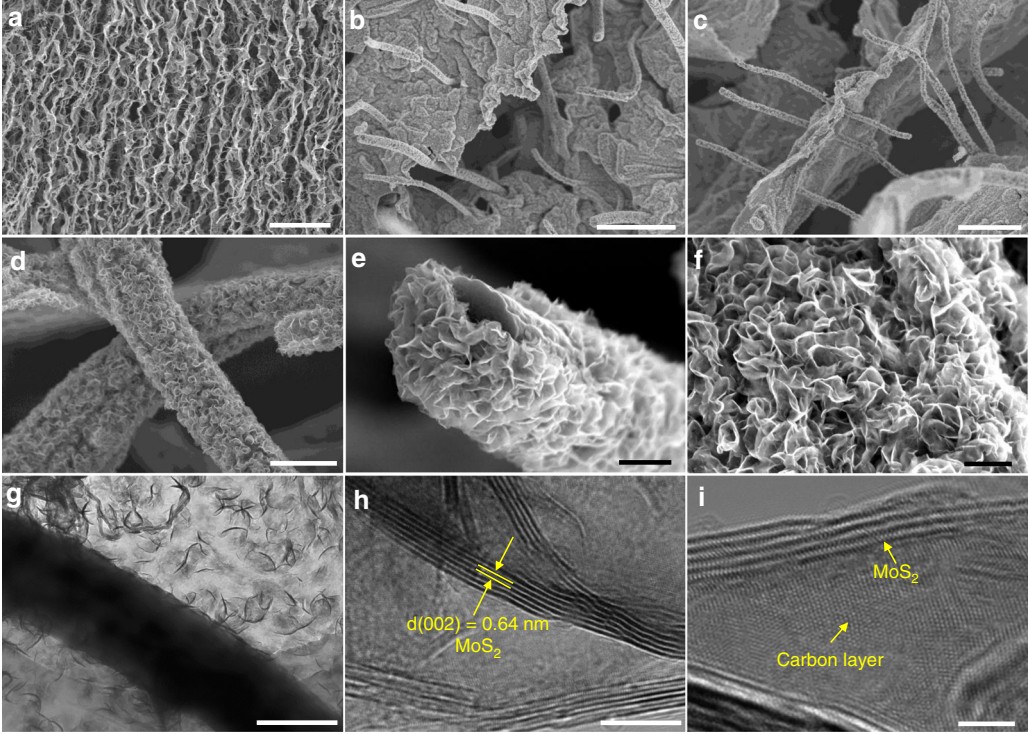

**Fig. 4** SEM analysis of the as-prepared MoS₂@CNFIG: SEM images of **a** overall structure (scale bar = 50 μm) and **b**, **c** cross section of MoS₂@CNFIG hybrid (scale bars, **b** 10 μm; **c** 5 μm); **d** MoS₂ layers homogeneously anchored on the CNF bridges (scale bar = 2 μm); **e** SEM image of an endpoint of CNF coated with MoS₂ nanoflakes (scale bar = 1 μm); **f** SEM image of selected area of MoS₂@CNFIG hybrid with MoS₂ nanoflakes anchored on the carbon layers (scale bar = 500 nm); **g** TEM analysis of the MoS₂@CNFIG sample (scale bar = 500 nm); **h**, **i** HRTEM images with distinctive layer spacing of MoS₂ and carbon matrix (scale bars, **h** 10 nm; **i** 2 nm)

cross section of the MoS₂@CNFIG hybrid further demonstrate that the CNFs are vertically aligned with the carbon layers (Fig. 4b, c). This result confirms the stable architecture with porous channels within the MoS₂@CNFIG hybrid. The homogeneously deposited MoS₂ skins on the surface of the carbon layers and the CNFs are clearly illustrated by the SEM image at high magnification (Fig. 4d). The MoS₂ nanomaterials possess good porous structures and "nanoflake" morphologies, contributing to a much higher specific surface area of MoS₂@CNFIG hybrid (310 mg⁻¹) compared with pure MoS₂ (92 mg⁻¹) (Supplementary Fig. 12), which can further permit complete utilization of their active sites during the charge/discharge process. In this work, the thickness of the MoS₂ skin is about 200 nm, which can be obtained from the endpoint of a CNF pillar coated by MoS₂ nanoflakes (Fig. 4e). Furthermore, the carbon layers are also completely wrapped by the homogeneously deposited MoS₂ layers (Fig. 4f). MoS₂@CNFIG hybrid aerogel can be cut into round pieces with an average thickness of 2.5 mm (Supplementary Fig. 13), which can be directly used as anode for LIBs. In this MoS₂@CNFIG hybrid aerogel, the large voids among the graphene layers can play two roles on the battery performance (i.e., specific capacity and rate capability). On one hand, as indicated by the examiner, the voids among the graphene layers could reduce the tap density and therefore lower the volumetric energy density. On the other hand, since it is widely recognized that sodium ion have a much larger size than lithium ion, the voids in the CNFIG architecture provide the space to grow MoS₂ nanoflakes, offer ample pathways for the mass transport and storage of large sodium ions, tolerate the volume changes during charge/discharge processes.

From these SEM observations, we notice that (i) the good porous structure of the CNFIG matrix is maintained, (ii) MoS₂

nanomaterials with 2D flake-like geometry are homogeneously deposited on the surface of CNFIG without any aggregation, and (iii) the thin MoS₂ layers are in tight contact with the CNFIG matrix which can greatly decrease the interfacial resistance for the MoS₂@CNFIG hybrid. These structural characteristics can facilitate the rapid transmission of ions and electrons contributing to the outstanding electrochemical performances of the MoS₂@CNFIG hybrid. In comparison, randomly arranged CNFs/graphene sheets/MoS₂ (CNF/G/MoS₂) composites without vertically aligned pores exhibit aggregated morphologies, as seen in Supplementary Fig. 14. Also, pure MoS₂ materials exhibit "sphere" morphologies with a diameter of 2 μm (Supplementary Fig. 15) wherein a large number of their active sites are closely wrapped inside the MoS₂ spheres, potentially unused during the electrochemical reaction processes.

Transmission electron microscopy (TEM) and high-resolution TEM (HRTEM) were used to further investigate the morphological features and crystal structures of the MoS₂@CNFIG hybrid. The TEM images of MoS₂@CNFIG at low magnification (Fig. 4g) demonstrate the successful hybridization of CNFs, carbon layers and MoS₂ nanoflakes. The MoS₂ nanomaterials with a 2D flake-like morphology are homogeneously anchored on the surface of CNFs and carbon layers without any aggregation. This suggests that the synthesis strategy developed in this work ensures a quasi-epitaxial growth of MoS₂ along the 1D CNFs and 2D carbon layers. The large number of pores (Supplementary Fig. 16a), permit exploitation of all the MoS₂ active sites during the sodium ion insertion/extraction process. Meanwhile, the semitransparent MoS₂ layers (Supplementary Fig. 16b) indicate that the anchored MoS₂ nanoflakes consist of only a few layers without severe restacking. The thickness of the MoS₂ layer is about or <200 nm (Supplementary Fig. 17), which is consistent with the SEM result

(Fig. 4e). The thin layer geometry of $MoS_2$ nanoflakes can be also confirmed by the HRTEM images (Fig. 4h, i), in which 4–8 layers of $MoS_2$ with an expanded $d$-spacing of 0.64 nm can be detected. The tight contact between carbon layers and the anchored $MoS_2$ nanoflakes is also confirmed by these HRTEM images. The carbon matrix with a typical $d$-spacing of 0.34 nm, corresponding to the (002) crystal phase, is homogeneously hybridized with the introduced $MoS_2$ nanoflakes.

Energy-dispersive X-ray (EDX) analysis was conducted to confirm the uniform deposition of $MoS_2$ on the carbonic CNFIG networks (Supplementary Fig. 18). The SEM image of the selected cross section of MoS₂@CNFIG is presented in Supplementary Fig. 18a and the corresponding EDX elemental distributions of C, S, and Mo elements are presented in Supplementary Fig. 18b–d, respectively. These elemental mappings indicate the successful hybridization of $MoS_2$ nanoflakes and the CNFIG networks with homogeneous distribution. Here, the poor C element signals can be ascribed to the carbon networks being completely enclosed by the homogeneously anchored $MoS_2$ nanoflakes. Supplementary Fig. 19 exhibits the XRD patterns of the CNFIG framework, pure $MoS_2$ and the MoS₂@CNFIG hybrid. The pure CNFIG matrix exhibits a broad diffraction peak at around 25.9°, which is related to the (002) crystal plane of the carbon materials[46]. The diffraction peaks of pure $MoS_2$ can be indexed to the hexagonal phase of $MoS_2$ material (JCPDS No. 37−1492). Similar diffraction peaks at 14.02°, 33.28°, and 58.52° detected in the XRD patterns of the MoS₂@CNFIG hybrid can be assigned to the (002), (100), and (110) planes of $MoS_2$ crystals[47,48]. Here, the diffraction peak at 14.02° corresponds to an interlayer spacing of 0.64 nm, which is a little larger than that of other reported $MoS_2$ materials (0.62 nm)[49]. Thus, the expanded $d$-spacing of $MoS_2$ in the MoS₂@CNFIG hybrid can efficiently improve the insertion/extraction kinetics of sodium ions. Thermogravimetric analysis (TGA) measurement was conducted to determine the weight percentage of $MoS_2$ materials in the developed MoS₂@CNFIG hybrid (Supplementary Fig. 20). The slight weight loss before 200 °C can be ascribed to water evaporation. The apparent decreasing curve between 240 and 405 °C indicates the oxidation of $MoS_2$ to $MoO_3$. The combustion of carbon matrix occurred between 405 and 520 °C, and the weight loss at temperatures higher than 670 °C was due to the evaporation of $MoO_3$ in air[50]. Here, the weight percentage of $MoS_2$ in the MoS₂@CNFIG hybrid is calculated to be approximately 88.0 wt%.

**Electrochemical properties**. The electrochemical properties of the MoS₂@CNFIG hybrid and pure $MoS_2$ were evaluated by assembled coin cells with pure sodium metal as the counter electrode, and pure $MoS_2$ or MoS₂@CNFIG hybrid as the anode materials (Fig. 5). Figure 5a presents the cyclic voltammograms (CVs) of the MoS₂@CNFIG hybrid at 0.1 mV s$^{-1}$ in the 1st, 2nd, and 5th cycles between 0.1 and 3.0 V. The reduction process of $MoS_2$ can be divided into two steps: (i) the insertion of sodium ions into $MoS_2$ interlayers (Eq. 1) and (ii) the conversion of $MoS_2$ to Mo accompanied by the formation of $Na_2S$ (Eq. 2)[51,52].

$$MoS_2 + xNa^+ + xe^- = Na_xMoS_2 \qquad (1)$$

$$Na_xMoS_2 + (4-x)Na^+ + (4-x)e^- = 2Na_2S + Mo \qquad (2)$$

In the first cathodic scan, a strong peak observed at 0.6 V is associated with $Na^+$ insertion into the $MoS_2$ interlayer spacing according to Equation 1, and the formation of a solid electrolyte interface (SEI) layer owing to the decomposition of the electrolyte[53]. The peak under 0.5 V in the deep cathodic process can be assigned to the electrochemical decomposition of $MoS_2$ to

form metallic (Mo) nanograins and amorphous $Na_2S$ matrix according to Eq. 2[54]. Also, a broad anodic peak at 1.75 V observed in the first charging process, can be ascribed to the oxidation of Mo nanograins to $MoS_2$[55]. The CV curves in the 2nd and 5th cycles almost overlapped, suggesting high reversibility and good cycling stability of sodium ions storage in this MoS₂@CNFIG hybrid.

Discharge/charge curves for the 1st, 2nd, and 5th cycles at a constant current density of 0.1 A g$^{-1}$ are shown in Fig. 5b. The initial discharge curve possesses a long plateau between 0.5 and 1.0 V, which is consistent with the large cathodic peak at about 0.6 V in the first CV curve. The MoS₂@CNFIG hybrid exhibits a highly reversible specific capacity of 598 mAh g$^{-1}$ at 0.1 A g$^{-1}$ based on the total mass of $MoS_2$ and CNFIG matrix. A recovered charge capacity of 585 mAh g$^{-1}$ can also be observed, indicating a high Coulombic efficiency of ~97.8% in the 2nd cycle. The in situ growth of $MoS_2$ nanoflakes with intimate contact between $MoS_2$ nanoflakes and the CNFIG matrix has effectively avoided irreversible capacity. The good reversible ability of sodium ions in this MoS₂@CNFIG hybrid is further confirmed by the superstable discharge/charge plateaus in the 2nd and 5th cycles (Fig. 5b). However, the coarse CV curves of pure $MoS_2$ (Supplementary Fig. 21) with much lower current intensities indicate inferior sodium ions storage capabilities. Also, a much lower specific capacity of 253 mAh g$^{-1}$ for pure $MoS_2$ is observed from its discharge/charge curves (Supplementary Fig. 22). The obvious decrease in the specific capacities in the 2nd (219 mAh g$^{-1}$) and 5th (179 mAh g$^{-1}$) cycles for pure $MoS_2$ further confirms its poor reversible capability for sodium ions storage.

The rate capacities can be used to demonstrate the sodium ion storage capabilities of the MoS₂@CNFIG hybrid at low and high current densities. The MoS₂@CNFIG hybrid exhibits reversible capacities of 594, 533, 498, 477, and 456 mAh g$^{-1}$ at current densities of 0.1, 0.5, 1.0, 2.0, and 5.0 A g$^{-1}$, respectively (Fig. 5c). More than 77% of the specific capacity observed at 0.1 A g$^{-1}$ is maintained when the current density is increased to 5.0 A g$^{-1}$, indicating that the MoS₂@CNFIG hybrid is a good anodic candidate at low and high current densities. More importantly, a high specific capacity of 582 mAh g$^{-1}$ can be achieved when the current density is returned to 0.1 A g$^{-1}$ after being cycled at high rates. This superior rate performance results from the excellent structural stability with the assistance of the inserted CNF pillars, and the intimate contact between $MoS_2$ and the conductive CNFIG matrix with greatly shortened sodium ion diffusion distances. The Coulombic efficiencies of the MoS₂@CNFIG hybrid gradually increase after the initial capacity loss, and quickly approach 100% after several cycles, indicating good reversibility. Comparatively, pure $MoS_2$ spheres exhibit a specific capacity of 253 mAh g$^{-1}$ at 0.1 A g$^{-1}$ and a much lower capacity of 32 mAh g$^{-1}$ when the current density is increased to 5.0 A g$^{-1}$. This inferior rate performance of pure $MoS_2$ can be ascribed to its heavy aggregation morphology that results in serious structural collapse after several cycles. The superior rate capability of the MoS₂@CNFIG hybrid can be further confirmed by the stable discharge/charge voltage profiles at 0.1, 0.5, 1.0, 2.0, and 5.0 A g$^{-1}$ (Fig. 5d).

The electrochemical impedance spectra (EIS) of the MoS₂@CNFIG hybrid and pure $MoS_2$ are presented in Fig. 5e. All the impedance measurements are made at the fully discharged state after 10 cycles. The impedance spectrum of the MoS₂@CNFIG hybrid is composed of a depressed semicircle in high-medium frequencies and a straight line in low frequencies. The non-symmetrical semicircle at high-medium frequencies consists of two parts, the resistance of SEI film ($R_s$) and charge transfer resistance ($R_{ct}$)[56]. The sloping line in the low frequency is associated with the diffusion kinetics of $Na^+$ in active materials[53].

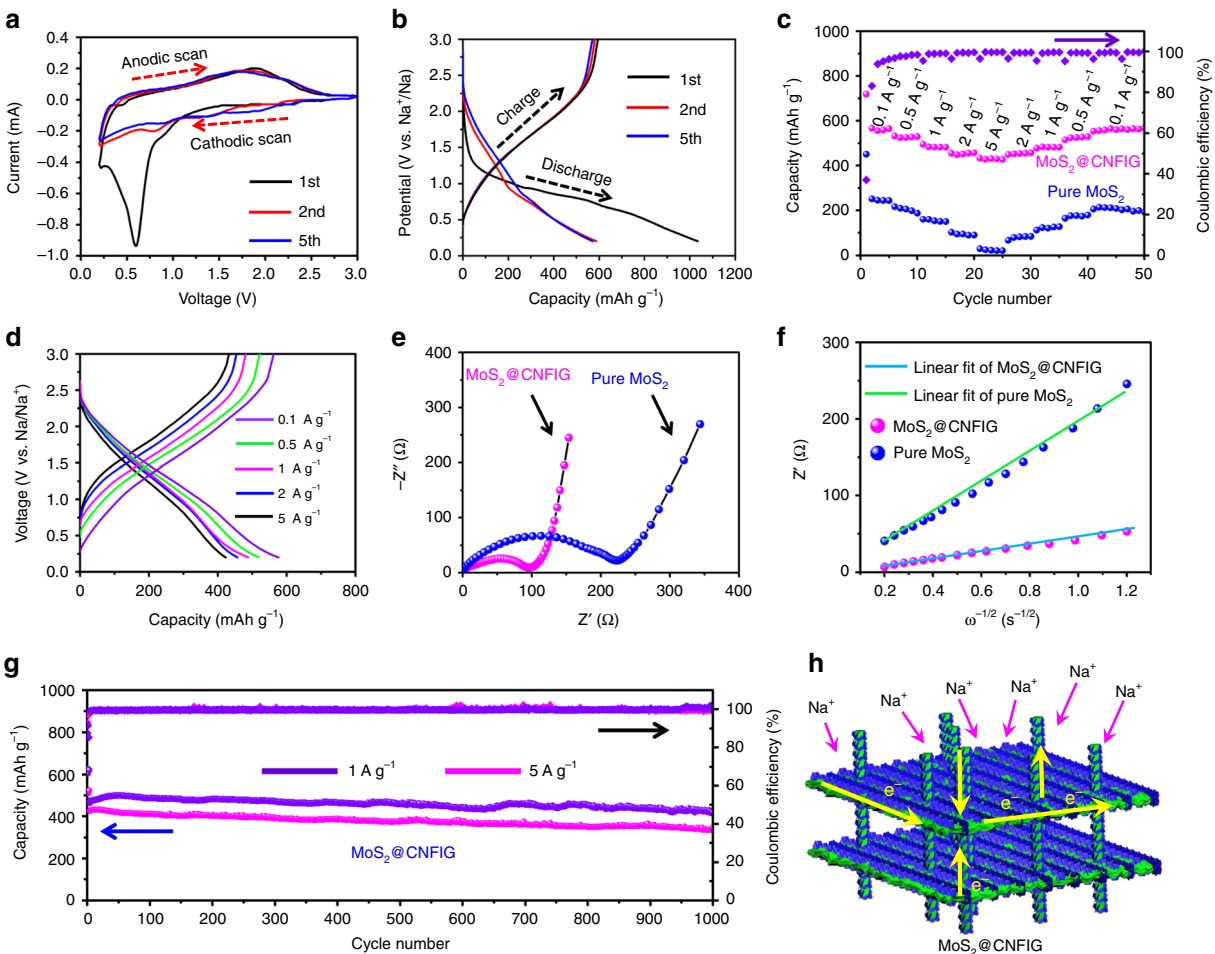

**Fig. 5** Electrochemical performances of MoS$_2$@CNFIG hybrid and pure MoS$_2$. **a** CV curves at 0.1 mV s$^{-1}$ and **b** discharge/charge curves at 0.1 A g$^{-1}$ of MoS$_2$@CNFIG in the 1st, 2nd, and 5th cycles; **c** rate performances of pure MoS$_2$ and MoS$_2$@CNFIG hybrid from 0.1 to 5 A g$^{-1}$; **d** discharge/charge curves of MoS$_2$@CNFIG hybrid at different current densities; **e** Nyquist plots of the pure MoS$_2$ and MoS$_2$@CNFIG after being cycled 5 times at 0.1 A g$^{-1}$; **f** the Z′–ω $^{-1/2}$ curves in the low-frequency region of Nyquist plots for pure MoS$_2$ and MoS$_2$@CNFIG hybrid, respectively; **g** long-term cycling performances of MoS$_2$@CNFIG at 1.0 and 5.0 A g$^{-1}$ and their corresponding Coulombic efficiencies; **h** schematic illustration of the Na$^+$ storage mechanism and electronic conductivity in the MoS$_2$@CNFIG hybrid with stable "interpenetrating networks"

Here, the kinetic parameters can be obtained from the equivalent circuit (Supplementary Fig. 23) which was utilized for fitting the EIS spectra. The MoS$_2$@CNFIG hybrid exhibits a much lower $R_{ct}$ value (100 Ω) than that of pure MoS$_2$ (234 Ω), indicating its higher electronic/ionic conductivity. Here, the Warburg impedance data in the low-frequency region of the Nyquist plots are utilized to analyze the chemical diffusion coefficient of sodium ions. Figure 5f shows the fitted line of Z′–ω$^{-1/2}$ ($\omega = 2\pi f$) in the low frequencies. The lower slope ($\sigma = 33.4$) of the MoS$_2$@CNFIG hybrid compared to pure MoS$_2$ ($\sigma = 242.3$) indicates the superior insertion/extraction kinetics of sodium ions in the MoS$_2$@CNFIG electrode.

Long-term cycling behaviors of the MoS$_2$@CNFIG hybrid are shown in Fig. 5g. Here, coin cells with MoS$_2$@CNFIG anodes exhibit excellent long-term cycling stability at 1 A g$^{-1}$. The specific capacities of the MoS$_2$@CNFIG hybrid are slightly increased in the first 40 cycles due to the chemical activation of the MoS$_2$ nanoflake active sites in the MoS$_2$@CNFIG hybrid. The MoS$_2$@CNFIG hybrid exhibits a sustainable specific capacity of 412 mAh g$^{-1}$ in the 1000th cycle, delivering capacity retention of 86.2% based on its initial specific capacity (478 mAh g$^{-1}$) in the 2nd cycle. More importantly, the MoS$_2$@CNFIG hybrid also exhibits a promising cycling life even at high current density of 5 A g$^{-1}$. A reversible specific capacity of 366 mAh g$^{-1}$ was

achieved at 5 A g$^{-1}$ after 1000 cycles, achieving capacity retention of 86.9%. The MoS$_2$@CNFIG hybrid also achieves high Coulombic efficiencies approaching ~100% both at 1 and 5 A g$^{-1}$, indicating the excellent reversible insertion/extraction ability of sodium ions inside its interpenetration networks. Comparatively, pure MoS$_2$ exhibits a sharp capacity decrease in the first 100 cycles (Supplementary Fig. 24). A much lower specific capacity of 34 mAh g$^{-1}$ is obtained after 1000 cycles with poor Coulombic efficiencies. Herein, the excellent sodium ions storage properties of the MoS$_2$@CNFIG hybrid can be ascribed to its hierarchical geometry with aligned channels, high electronic/ionic conductivity and uniform dispersion of thin MoS$_2$ layers. The introduced CNFs acting as the supporting pillars are beneficial for maintaining the structural integrity of the MoS$_2$@CNFIG hybrid by suppressing the stacking of MoS$_2$ nanoflakes, resulting in its superior electrochemical stability. The excellent porous structures with stable channels and the perfect electronic/ionic conductivity achieved by the MoS$_2$@CNFIG hybrid ensure the reversible insertion/extraction of sodium ions at a large scale, which promotes the exploitation of all the active sites of MoS$_2$ nanoflakes in rapid charge/discharge processes.

A CNF/G/MoS$_2$ composite was also used an anode material to fabricate the sodium cell, to examine its electrochemical activity for sodium ions storage. This randomly arranged CNF/G/MoS$_2$

composite without any aligned channels exhibits comparable specific capacities (584 mAh g$^{-1}$) to that of the MoS$_2$@CNFIG hybrid at low current density of 0.1 A g$^{-1}$ (Supplementary Fig. 25). However, when the testing current density is increased to 5.0 A g$^{-1}$, the sodium ions storage capability of the CNF/G/ MoS$_2$ composite is severely decreased to 120 mAh g$^{-1}$, as confirmed by its rate performance (Supplementary Fig. 26). This poor performance of the CNF/G/MoS$_2$ composite can be ascribed to the absence of efficient aligned channels and the unexpected utilization of the non-conducting polymer binder of PVDF. Furthermore, this CNF/G/MoS$_2$ composite also shows an inferior long-term cycling life, with only 55% of its initial specific capacity maintained (Supplementary Fig. 27). Comparison of the electrochemical performances of the CNF/G/MoS$_2$ composite with those of the MoS$_2$@CNFIG hybrid further confirm the structural advantages of the vertically aligned channels and robust template matrix, which greatly contribute to the structural stability of the electrode and provide efficient pathways for the transfer of sodium ions and electrons. The EIS spectra of sodium ions cells with CNF/G/MoS$_2$ composite anodes are presented in Supplementary Fig. 28. A higher $R_{ct}$ value (166 Ω) of the CNF/G/MoS$_2$ composite demonstrates a larger charge transfer resistance. Here, the $R_{ct}$ value (166 Ω) of the CNF/G/MoS$_2$ composite is lower than the pure MoS$_2$ electrode (234 Ω), due to the introduced graphene sheets template that restricts the restacking of the MoS$_2$ nanoflakes, and the inserted CNFs that increase the internal electrical conductivity. The higher slope ($\sigma = 92.3$) of the CNF/G/ MoS$_2$ composite (Supplementary Fig. 29) compared with the MoS$_2$@CNFIG hybrid ($\sigma = 33.4$) demonstrates less efficient sodium ions transfer inside the CNF/G/MoS$_2$ composite anode, which limits rapid sodium ions insertion/extraction under high current densities. Moreover, the MoS$_2$@CNFIG hybrid is comparable with or superior to other types of carbon/MoS$_2$ composites (Supplementary Fig. 30).

The structural integrity of the MoS$_2$@CNFIG anode after being cycled 1000 times is further confirmed by SEM imagery (Fig. 6a). The vertically aligned pores of the MoS$_2$@CNFIG anode are completely preserved, which can ensure efficient diffusion pathways for the electrolyte even after a long-term cycling process, and provide sufficient expansion space for MoS$_2$ nanoflakes. All these features can undoubtedly contribute to the reversible insertion/extraction of sodium ions. EDX mapping of Na, Mo, and S elements can be clearly detected (Fig. 6b), confirming the stable structure of the MoS$_2$@CNFIG hybrid and its effective adsorption of sodium ions. The elemental map of Na can be ascribed to the sodium ions being adsorbed via chemical redox reactions. The mapping signal of C is due to the complete wrapping of the carbon shells by MoS$_2$ nanoflakes. This result demonstrates that the anchored MoS$_2$ nanoflakes do not fall off, even after long-term cycling, and the tight contact between MoS$_2$ nanoflakes and the carbon shells. Figure 6c shows that 12 light-emitting diodes (LEDs) can be lit up by three coin cells connected in series based on MoS$_2$@CNFIG anodes, which further confirms the potential practical applications of the developed MoS$_2$@CN-FIG hybrid.

To evaluate the superior long-term cycling capability coupled with a good rate performance, the MoS$_2$@CNFIG hybrid was first cycled at 1 A g$^{-1}$ for 300 cycles then continuously cycled at high current density up to 10 A g$^{-1}$ for 400 cycles (Fig. 6d). The MoS$_2$@CNFIG hybrid exhibits a stable cycling performance by achieving a reversible specific capacity of 322 mAh g$^{-1}$ in the 300th cycle at 10 A g$^{-1}$ and 303 mAh g$^{-1}$ in the 700th cycle. When the current density is returned to 1 A g$^{-1}$, the MoS$_2$@CN-FIG hybrid shows a recoverable specific capacity of 421 mAh g$^{-1}$ and can be further cycled 300 times with high Coulombic efficiencies approaching ~100%. Here, the superior rate

performances of MoS$_2$@CNFIG hybrid can be ascribed to its excellent structural stability and the ultra-high utilization efficiency of MoS$_2$ nanoflakes due to their structural features. The EIS spectra of the MoS$_2$@CNFIG hybrid after being cycled two and 1000 times are presented in Fig. 6e. The increased $R_{ct}$ value in the 1000th cycle (162.3 Ω) compared with the initial result in the 2nd cycle (100 Ω), further confirms the structural integrity and stable interfacial reaction during the long-term cycling process. The TEM image of the cycled MoS$_2$@CNFIG hybrid exhibits flake-like MoS$_2$ structures (Fig. 6f) with the existence of clear lattice fringes (inset in Fig. 6f). Here, the promising electrochemical storage of sodium ions can be ascribed to the hierarchical structure of the MoS$_2$@CNFIG hybrid (Fig. 6g). Firstly, the nested structures produce nanoreservoirs between adjacent MoS$_2$ nanoflakes, which favor interfacial interactions between the active sites of MoS$_2$ and the electrolyte, and shorten the ionic diffusion pathways[57]. Secondly, the vertically aligned channels ensure the rapid penetration of electrolyte and sodium ions, which helps relieve the mass-transfer limitations of the electrochemical MoS$_2$-sodium ions reactions[58]. Thirdly, the tightly anchored MoS$_2$ on the surface of the CNFIG matrix provides good current collector/MoS$_2$ electrical contact and much lower $R_{ct}$ resistance. Lastly, the vertically aligned channels and the porous structure of the MoS$_2$@CNFIG hybrid provide sufficient volume expansion space for active MoS$_2$ nanomaterials (Fig. 6g), avoiding the structural collapse responsible for irreversible capacities.

## Discussion

In summary, we have rationally designed and successfully fabricated a 3D MoS$_2$@CNFIG nanohybrid with unique interpenetration networks as a free-standing anode for SIBs (without the use of conductive additives and binders). The as-prepared CNFIG framework provides ultra-stable channels and the supporting pillars between different carbon layers, facilitating efficient pathways for the rapid penetration of electrolyte and quick transfer of sodium ions. The excellent electrical conductivity of the CNFIG matrix, coupled with the robust interfacial contact between MoS$_2$ nanoflakes and CNFIG matrix, enable low charge transfer resistance and full utilization of active sites of the anchored MoS$_2$ electroactive materials. The ultra-high mechanical compression property of the CNFIG matrix can contribute to the structural stability of the MoS$_2$@CNFIG hybrid, avoiding any unexpected structural collapse from the volume expansion of MoS$_2$ materials in charge/discharge processes. As a result, a high specific capacity of 598 mAh g$^{-1}$ and a long-term cycling stability up to 1000 times with an average Coulombic efficiency of ~100% are achieved by this MoS$_2$@CNFIG hybrid. Importantly, the MoS$_2$@CNFIG hybrid also possesses an excellent rate performance even at a high current density up to 10 A g$^{-1}$ due to its unique interpenetration networks. Furthermore, this MoS$_2$@CNFIG hybrid provides new insights for designing and fabricating good porous electrode materials for energy storage in other fields with high capacity, long cycling life, and excellent rate performances.

## Methods

**Materials.** Concentrated sulfuric acid (H$_2$SO$_4$, 98%), thiourea (CH$_4$N$_2$S), hex-ammonium molybdate ((NH$_4$)$_6$Mo$_7$O$_{24}$), N, N-Dimethylacetamide (DMAc), triethylamine (TEA), 4,4′-oxidianiline (ODA), and pyromellitic dianhydride (PMDA) were purchased from Shanghai Chemical Reagent Company. Deionized (DI) water was used throughout the experiments.

**Synthesis of PAA powders.** Poly(amic acid) (PAA) powders were prepared based on ODA and PMDA. Typically, 2.15 g of ODA was dispersed into 27.5 mL of DMAc by strong stirring at 0 °C. Then, PMDA (2.35 g) was gradually added into the mixture and the reaction was maintained for 5 h in an ice bath. TEA (1.1 g) was

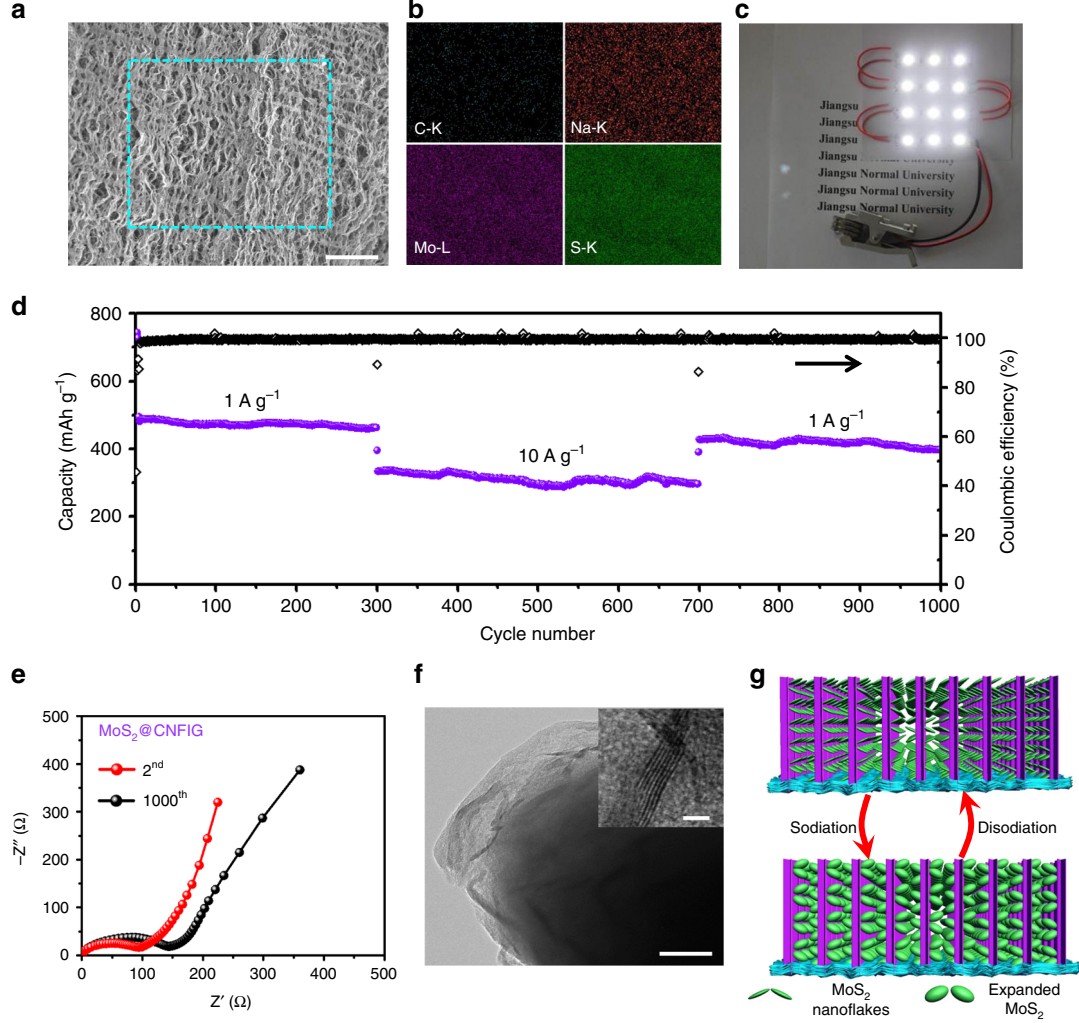

**Fig. 6** The structural evolution and electrochemical performances of MoS₂@CNFIG anode. **a** SEM image of MoS₂@CNFIG anode after being cycled 1000 times (scale bar = 50 μm) and **b** corresponding EDX elemental mappings of C, Na, Mo, and S; **c** twelve LEDs can be lit up by three coin cells connected in series based on the as-prepared MoS₂@CNFIG anodes; **d** long-term cycling performance of MoS₂@CNFIG at 1 and 10 A g⁻¹; **e** EIS spectra of MoS₂@CNFIG hybrid after being cycled two and 1000 times; **f** TEM image of the MoS₂@CNFIG hybrid cycled 1000 times (scale bar = 50 nm; Scale bar for inset in **f**, 2 nm); **g** schematic illustration of the tight contact between MoS₂ nanoflakes and the carbon shells, as well as the good porous space provided by the MoS₂@CNFIG hybrid for the volume expansion of MoS₂ nanoflakes

dropped into the mixture drop-by-drop. After 5 h, a yellow viscous solution TEA-PAA, was obtained and poured into DI water. The precipitate was washed several times by DI water then freeze-dried, resulting in the formation of yellow PAA powders.

**Synthesis of CNF.** The reaction product based on ODA and PMDA prior to the addition of TEA can be directly used as a polymer matrix for electrospinning with a weight percent of ~15%. Electrospinning was carried out at an applied voltage of 12–18 kV with a feeding speed of 0.5 mL h⁻¹ with a distance of 18 cm between the syringe and the aluminum collector. The obtained PAA film was carbonized at 300 °C in air for 2 h and at 900 °C in Ar for 5 h, resulting in the formation of carbon nanofiber film. The obtained carbon nanofiber film was immersed into H₂SO₄ for 3 days, and further treated with strong sonication for 60 min following by washing with DI water, producing CNFs. For short CNFs with an average length of 2–3 μm, the obtained long CNFs were further treated with ball-milling at 400 rpm for 4 h.

**Preparation of CNFIG.** GO materials were prepared according to a modified Hummers' method[39]. CNFs (100 mg) were dispersed into a GO solution (200 mL, 1 mg mL⁻¹) under strong stirring and sonication. Then, PAA powder (0.4 g) was dispersed into the mixture with the assistance of TEA (2 mL). The obtained mixture was vertically frozen with the bottom of the container gradually immersed into liquid nitrogen. The frozen GO/CNF/PAA composite was freeze-dried at −50 °C under 10 Pa, and the obtained GO/CNF/PAA aerogel was

further carbonized at 900 °C in Ar for 5 h, resulting in the formation of CNF-interpenetrated graphene, named as CNFIG.

**Preparation of MoS₂@CNFIG.** (NH₄)₆Mo₇O₂₄ (1.5 mmol) and CH₄N₂S (21 mmol) were dissolved into 40 ml of ultrapure water, and 200 mg of CNFIG cake were put into the solution. The resultant mixture was subjected to a hydrothermal reaction at 220 °C for 12 h in a Teflon-lined stainless steel autoclave (50 mL). The obtained dry solid materials were high-temperature treated at 700 °C for 2 h, resulting in **MoS₂@CNFIG** active materials. Pure MoS₂ materials were prepared according to the same method without the addition of the **CNFIG** matrix.

**Preparation of CNF/G/MoS₂.** CNFs (100 mg), GO sheets (200 mg) and PAA matrix (0.4 g) were firstly co-dispersed with the assistance of strong sonication (1000 W, 40 KHz). And these CNFs/GO/PAA composite were dried in the oven at 80 °C overnight, and were further carbonized in the furnace tube at 900 °C for 5 h in Ar, achieving the development of CNFs/graphene sheets (CNF/G) composite powder. Then, (NH₄)₆Mo₇O₂₄ (1.5 mmol) and CH₄N₂S (21 mmol) were dissolved into 40 mL ultrapure water, and 200 mg of CNF/G powder were dispersed into the above solution. Then the resultant mixture was subjected to a Teflon-lined stainless steel autoclave (50 mL) and further reacted at 220 °C for 12 h. The obtained dry solid materials were treated at 700 °C for 2 h, resulting in CNF/G/MoS₂ active materials.

**Materials characterization**. The nanostructures and morphologies of the prepared IN-C matrix and MoS$_2$@CNFIG hybrid were studied by field-emission scanning electron microscopy (SEM, Hitachi, SU8010) and transmission electron microscopy (TEM, FEI Tecnai G2 F20). Energy dispersive X-ray spectroscopy (EDX) detections were captured with an EDAX (PW9900). The weight percent of samples were determined using thermogravimetric analysis (TGA) equipment (TA 500) from room temperature to 800 °C with a heating rate of 10 °C min$^{-1}$. The crystalline phases of the developed products were characterized by a XRD diffractometer (Bruker. D8 Advanced) with Cu K$\alpha$ = 0.154056 nm. The compression tests of MoS$_2$@CNFIG hybrid were performed on an electronic universal testing machine (SANS, CMT6103). Electrical conductivity was tested on an electrochemical workstation (CHI 660D).

**Electrochemical measurements**. Electrochemical measurements of the prepared materials were carried out by two-electrode CR2032 coin-type cells. The MoS$_2$@CNFIG hybrid was used as binder-free anodes, and sodium foil was used as the cathode with a microporous glass fiber separator (Whatman) placed between the sodium metal counter electrode and the working electrode. 1 M NaClO$_4$ (Alfa Aesar) in (1:1 v/v) dimethyl carbonate/ethylene carbonate was used as the electrolyte. A washer, spring, and top casing were placed on top to complete the assembly before crimping. For CNF/G/MoS$_2$ and pure MoS$_2$ anodes, they were mixed with conductive additions and polymer binder, and were further coated on copper foil to form the anode electrodes. For example, CNF/G/MoS$_2$ (or pure MoS$_2$) powders (80 wt%) were mixed with acetylene black (Super P, 10 wt%) and polyvinylidene fluoride (PVDF, 10 wt%) to prepare the pure MoS$_2$ working electrodes. Cyclic voltammograms (CVs) curves of the assembled coins were tested on a BT2000 ARBIN between 0.1 and 3.0 V vs. Na/Na$^+$. Discharge/charge curves of the assembled coins were recorded on LAND 2001A testing systems. Electrochemical impedance spectroscopy (EIS) measurements were carried out based on a Princeton-solartron system over the frequency range 100 kHz to 0.01 Hz under an open circuit potential. Here, it is necessary to declare that the calculated specific capacities of the prepared samples were based on the total mass of the MoS$_2$@CNFIG hybrid. The diffusion coefficient (D) of sodium ions inside the electrodes was calculated based on the EIS spectra in the low-frequency region according to the following equations:

$$D = \frac{R^2 T^2}{2A^2 n^4 F^4 C^2 \sigma^2}$$

$$Z_W = R_D + R_L + \sigma\omega^{-1/2}$$

Where $R$ is the gas constant, $T$ is the absolute temperature, A is the electrode area, n is the number of electrons per molecule during oxidization (for sodium ions, value is 1), $F$ is the Faraday constant, $C$ is the initial concentration (mol cm$^{-3}$) and $\sigma$ represents the Warburg factor, which is relative to $Zw$ according to the second equation above. The Warburg factor ($\sigma$) can be detected from the slope value based on $Zw$ with the square root of the frequency ($\omega^{-1/2}$).

## Data availability
The data that support the findings of this study are available from the corresponding author upon request.

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

## Acknowledgements

This work was financially supported by the National Natural Science Foundation of China (51703087, 51433001,51572116 and 51871113), and the Natural Science Foundation of Jiangsu Province (BK20150238, BK20170240).

## Author contributions

M.L., Y.Y., C.L., T.L. and S.Z. proposed and supervised the project. M.L., Y.Y. and P.Z. conceived and performed the experiments. M.L. and P.Z. synthesized the CNFIG materials. Z.Q. did the SEM measurements. Y.Y. and P.Z. fabricated the batteries and carried out the electrochemical performance tests. M.L., Y.Y., C.L., T.L. and S.Z. wrote the paper. All the authors contributed to the results analyses, discussions, and have approved the final version of the paper.

## Additional information

**Competing interests:** The authors declare no competing interests.

