## [Peer Review File · Nature Communications]

Reviewers' comments:

Reviewer #1 (Remarks to the Author):

In this manuscript, a robust 3D conductive carbon nanofibers (CNFs) interpenetrated graphene (CNFIG) architecture by directing CNFs to penetrate through the graphene sheets was designed. Then, MoS₂ nanoflakes were in-situ deposited on the surface of the CNFIG framework, producing a MoS₂@CNFIG hybrid. As a result, the as-prepared SIB delivers outstanding electrochemical performance and ultrahigh stability. This work is quite interesting and the floors-and-pillars concept in construction is proved to be effective. Therefore, the publication of this manuscript can be considered after major revisions:

1. The preparation of CNFIG is successful. However, the void among the graphene layers is significantly large, lowering the volumetric energy density of the composite. Thus, it is doubtful that the demonstration of the advantage of such structure design by the fabrication of electrodes for SIBs is appropriate.
2. The content of MoS₂ in the MoS₂@CNFIG composite was determined to be 93.3% by the TG analysis. Such value seems to be unreasonable. The authors should give details how to calculate the weight percentage of MoS₂. In addition, carbon cannot survive high temperature calcination. However, no weight loss was attributed to the decomposition of CNF and GO. Please give an explanation. In addition, is it possible to control the content of MoS₂ in the composite?
3. In the experimental, the authors mentioned that binder-free electrode was made by MoS₂@CNFIG. How to the binder-free electrode? What are the thickness and size of such electrode? Details should be provided.
4. The description of Fig. 1c ("Vertically aligned channels can be clearly observed in the overall image of the carbon networks") (line 104-105) does not match with the SEM image in the Fig. c (line 116, graphene sheets).
5. The authors give a background introduction about graphene in line 72-80, define "3D conductive carbon nanofibers (CNFs) interpenetrated graphene" as CNFIG (line 83), and give an SEM image of graphene in Figure 1c. But they use graphene oxide in figure 1a and the following manuscript. Graphene sheets and graphene oxide are totally different materials.
6. The author prepared the CNFs from the carbon fiber networks (Fig. S2) by ball-milling (line 100). But in the preparation section, CNFs were produced by strong sonication (line 480). In addition, as the authors mentioned, the length of CNFs plays a key role in the preparation of CNFIG architectures (line 176). Then how did you control the length of the used CNTs?
7. What is CNF/G/MoS₂? How did you synthesize it? And how did you prepare the electrode of CNF/G/MoS₂? What is the content of MoS₂ in this composite? Please provide the details in order to avoid confusion

Reviewer #2 (Remarks to the Author):

Recommendation: minor revision

Comments:

This paper reports the synthesis of 3D conductive CNFIG architecture, in which CNFs were vertically penetrated through the carbon layers. A compressive stress of 0.25 MPa has been achieved due to the effective supporting of the vertically aligned CNFs. This architecture is very interesting, and the synthesis process is facile-yet-efficient. And the formation mechanism of this structure has been analyzed and presented. Moreover, this architecture can be used as a good framework for loading MoS₂ nanoflakes, SEM images have confirmed the uniform distribution of the deposited MoS₂ layer. The

assembled sodium ion battery with this MoS₂@CNFIG anode delivers a remarkable specific capacity of 598 mAh/g with a long-term cycling stability of 1000 cycles. The academic thoughts in this work can provide new sights for designing and developing framework matrix for further construct novel hybrid electrodes for energy storage. The overall work is fancy and the whole paper is well organized. In this case, I believe this work can attract wide readers especially for energy storage researchers and is deserved to be published on Nature Communications after solving the following concerns.

1. In this work, the PAA powder has been firstly prepared and plays an important role in the development of CNFIG, but the authors have not provided the SEM or digital photo of this precursor.
2. Different lengths of CNFs have been tested in the formation of CNFIG. Whether the diameter of this CNF has large effect of this CNFIG architecture?
3. Whether other carbon materials, such as carbon nanotubes, can be vertically aligned through different carbon layers?
4. The authors have stated "The MoS₂ nanomaterials possess good porous structures and "nanoflake" morphologies...", BET results of these materials should be provided.
5. The authors have said "...and maintain exceptional mechanical integrity and excellent electrical conductivity during sodiation/disodiation processes", please provide the conductive data or confirmation of CNFIG.
6. Why the XRD result of MoS₂@CNFIG didn't present any diffraction peak of carbon matrix?

Reviewers' comments:

Reviewer #1 (Remarks to the Author):

In this manuscript, a robust 3D conductive carbon nanofibers (CNFs) interpenetrated graphene (CNFIG) architecture by directing CNFs to penetrate through the graphene sheets was designed. Then, MoS₂ nanoflakes were in-situ deposited on the surface of the CNFIG framework, producing a MoS₂@CNFIG hybrid. As a result, the as-prepared SIB delivers outstanding electrochemical performance and ultrahigh stability. This work is quite interesting and the floors-and-pillars concept in construction is proved to be effective. Therefore, the publication of this manuscript can be considered after major revisions:

1. The preparation of CNFIG is successful. However, the void among the graphene layers is significantly large, lowering the volumetric energy density of the composite. Thus, it is doubt that the demonstration of the advantage of such structure design by the fabrication of electrodes for SIBs is appropriate.

Response: Thanks for the positive comments on the successful preparation of CNFIG structure. The voids among the graphene layers play two roles on the battery performance (i.e., specific capacity and rate capability). On one hand, as indicated by the examiner, the voids among the graphene layers could reduce the tap density and therefore lower the volumetric energy density. On the other hand, since it is widely recognized that sodium ion have a much larger size than lithium ion, the voids in the CNFIG architecture provide the space to grow MoS₂ nanoflakes, offer ample pathways for the mass transport and storage of large sodium ions, tolerate the volume changes during charge/discharge processes. As a result, high discharge capacity and stable cycle performance (up to 1000 cycles) can be obtained even at high current density of 10 A g⁻¹. Greatly enhanced specific capacity, especially at high current density, can compensate the energy density reduction caused by the large voids. The above discussion is now added into the manuscript on Page 7.

2. The content of MoS₂ in the MoS₂@CNFIG composite was determined to 93.3% by the TG analysis. Such value seems to be unreasonable. The authors should give details how to calculate the weight percentage of MoS₂. In addition, carbon cannot survive high temperature calcination. However, No weight loss was attributed to the decomposition of CNF and GO. Please give an explanation. In addition, is it possible to control the content of MoS₂ in the composite?

Response: Thanks for the reviewer's sharp observation and constructive suggestion. As suggested, we conducted new experiments to accurately determine the weight percentage of MoS₂. Three kinds of samples including pure MoS₂, CNFIG matrix and MoS₂@CNFIG composite were retested by TGA (Figure S20), and the weight percentages of MoS₂ are calculated based on TGA curves and the following equation:

$$W_{MoS_2} \cdot x_{MoS_2} + W_{CNFIG} \cdot (1 - x_{MoS_2}) = W_{MoS_2@CNFIG}$$

where x_{MoS_2} is assigned as the weight percentage of MoS₂ in the MoS₂@CNFIG

composite. W_{MoS_2} , W_{CNFIG} , $W_{\text{MoS}_2@\text{CNFIG}}$ are the residual weight percentages of pure MoS₂, CNFIG matrix and MoS₂@CNFIG composite at 640 °C. The value of W_{MoS_2} , W_{CNFIG} , $W_{\text{MoS}_2@\text{CNFIG}}$ calculated based on the TGA curves are 81.4 wt%, 0.27 wt% and 71.7 wt%. According to the calculation, the weight percent of MoS₂ nanoflakes deposited on the CNFIG matrix is about 88.0 %. This corrected value is now added in the revised manuscript (highlight on page 9). The details of these experiments are given in Figure S20, and more discussion has been added in the revised manuscript.

Figure S20. TGA curves of pure MoS₂, CNFIG matrix and MoS₂@CNFIG composite in air at a heating rate of 10 °C min⁻¹ from room temperature to 800 °C.

In addition, the content of MoS₂ in this MoS₂@CNFIG composite was determined by the inserted precursors of (NH₄)₆Mo₇O₂₄ and CH₄N₂S. Thus, it is possible and facile to control the MoS₂ content by adjusting the amount of precursors in the synthesis process.

3. In the experimental, the authors mentioned that binder-free electrode was made by MoS₂@CNFIG. How to the binder-free electrode? What are the thickness and size of such electrode? Details should be provided.

Response: Traditional electroactive materials are commonly in powder form. The fabrication of electrode inevitably involve the used of polymer binder (such as PVDF, PTFE, and so on) and conducting additive (e.g., carbon black) in order to immobilize the electroactive materials onto current collector (i.e., copper or aluminum foil). In this work, the developed MoS₂@CNFIG composite has already incorporated the electroactive materials (MoS₂) and current collecting network (i.e., CNFIG) together as a free-standing electrode, removing the need of adding any polymer binder and carbon black. The pristine thickness of this MoS₂@CNFIG anode is about 2.5 mm with a diameter of about 12 mm, as seen in Figure S13, and it can be pressed before being used as an anode for SIB.

4. The description of Fig. 1c (“Vertically aligned channels can be clearly observed in the overall image of the carbon networks”) (line 104-105) does not match with the

SEM image in the Fig. c (line 116, graphene sheets).

Response: We are sorry for making a mistake for the figure order. The description of Fig. 2c (graphene oxide sheets) is now added in the revised manuscript, and the original Fig. 1c is corrected as Fig. 1d.

5. The authors give a background introduction about graphene in line 72-80, define “3D conductive carbon nanofibers (CNFs) interpenetrated graphene” as CNFIG (line 83), and give an SEM image of graphene in Figure 1c. But they use graphene oxide in figure 1a and the following manuscript. Graphene sheets and graphene oxide are totally different materials.

Response: Thank you for your suggestion. In this work, graphene oxide was used as raw 2D material for fabricating the CNFIG. However, the graphene oxide sheets were converted to graphene sheets after being thermally treated at 900 °C in Ar. That is, in the final products, the graphene oxide sheets have been converted to graphene sheets after high-temperature treatment. For clarification, we use graphene oxide (GO) in the raw materials and graphene sheets (G) in the final products in the revised manuscript.

6. The author prepared the CNFs from the carbon fiber networks (Fig. S2) by ball-milling (line 100). But in the preparation section, CNFs were produced by strong sonication (line 480). In addition, as the authors mentioned, the length of CNFs plays a key role in the preparation of CNFIG architectures (line 176). Then how did you control the length of the used CNTs?

Response: Thank you very much for this comment. In this work, the CNFs with an average length of 30-40 μm were prepared by the sonication treatment in a concentration H_2SO_4 solution. While, the short CNFs with length of 2-3 μm were obtained by an additional ball-milling process. Detailed procedures are now provided in the experimental section (highlight on page 15).

7. What is CNF/G/MoS₂? How did you synthesize it? And how did you prepare the electrode of CNF/G/MoS₂? What is the content of MoS₂ in this composite? Please provide the details in order to avoid confusion

Response: In this work, CNF/G/MoS₂ is a control sample, representing the simple mixture of CNF, Graphene and MoS₂ (without the vertically frozen treatment). These CNFs/graphene sheets/PAA materials were dried in the oven, and the same amount of (NH₄)₆Mo₇O₂₄ (1.5 mmol) and CH₄N₂S (21 mmol) were used for the deposition of MoS₂ nanoflakes in the hydrothermal reaction, in order to ensure the MoS₂ content in this CNF/G/MoS₂ materials is the same as that in the MoS₂@CNFIG composite. For CNF/G/MoS₂ and pure MoS₂ anodes, they were mixed with the conductive additives and polymer binder, and coated on copper foil to form the anode electrodes. In particular, CNF/G/MoS₂ (or pure MoS₂) powders (80 wt%) were mixed with acetylene black (Super P, 10 wt%) and polyvinylidene fluoride (PVDF, 10 wt%) to prepare the pure MoS₂ working electrodes. All these details were added in the experimental section (highlight on page 16).

Reviewer #2 (Remarks to the Author):

Recommendation: minor revision

Comments:

This paper reports the synthesis of 3D conductive CNFIG architecture, in which CNFs were vertically penetrated through the carbon layers. A compressive stress of 0.25 MPa has been achieved due to the effective supporting of the vertically aligned CNFs. This architecture is very interesting, and the synthesis process is facile-yet-efficient. And the formation mechanism of this structure has been analyzed and present. Moreover, this architecture can be used as a good framework for loading MoS₂ nanoflakes, SEM images have confirmed the uniform distribution of the deposited MoS₂ layer. The assembled sodium ion battery with this MoS₂@CNFIG anode delivers a remarkable specific capacity of 598 mAh/g with a long-term cycling stability of 1000 cycles. The academic thoughts in this work can provide new sights for designing and developing framework matrix for further construct novel hybrid electrodes for energy storage. The overall work is fancy and the whole paper is well organized. In this case, I believe this work can attract wide readers especially for energy storage researchers and is deserved to be published on Nature Communications after solving the following concerns.

1. In this work, the PAA powder has been firstly prepared and plays an important role in the development of CNFIG, but the authors have not provided the SEM or digital photo of this precursor.

Response: Thanks for the positive comments on the uniqueness of our materials and performance of our sodium ion battery.

As suggested, the digital photo of the PAA powder is now provided as shown in Figure S5. The corresponding description of PAA powder is also added in the manuscript.

2. Different lengths of CNFs have been tested in the formation of CNFIG. Whether the diameter of this CNF has large effect of this CNFIG architecture?

Response: Yes, indeed the CNFs with other diameters can also be interpenetrated into the carbon layers. Here, we have successfully facilitate CNFs interpenetrate across the carbon layers of a larger diameter of 900 nm (see Figure S8). Corresponding descriptions are added in the revised manuscript.

3. Whether other carbon materials, such as carbon nanotubes, can be vertically aligned through different carbon layers?

Response: Following our proposed mechanism, we believe, carbon nanotubes, can also be vertically aligned through carbon layers.

4. The authors have stated “The MoS₂ nanomaterials possess good porous structures and “nanoflake” morphologies...”, BET results of these materials should be provided.

Response: Thanks for your suggestion. The specific surface area of MoS₂@CNFIG hybrid and pure MoS₂ materials were tested and provided, as shown in Figure S12. MoS₂@CNFIG hybrid shows a much higher surface area of 310 m²/g than 92 m²/g of MoS₂. Corresponding descriptions have been added in the revised manuscript

5. The authors have said “...and maintain exceptional mechanical integrity and excellent electrical conductivity during sodiation/disodiation processes”, please provide the conductive data or confirmation of CNFIG.

Response: As suggested by the reviewer, the electrical conductivity of CNFIG matrix was replenished, as seen in Figure S10, and an electrical conductivity of 15.6 S cm⁻¹ of CNFIG was achieved. Meanwhile, a digital photo showing CNFIG matrix substituting the copper wire in a turn-on electrical circle is presented to confirm the good electrical conductivity of CNFIG, as seen in Fig. S11. And corresponding descriptions have been added in the manuscript, as seen on page 6

6. Why the XRD result of MoS₂@CNFIG didn't present any diffraction peak of carbon matrix?

Response: In this work, the diffraction peak of carbon matrix is not significant in the XRD spectrum of MoS₂@CNFIG can be attributed to the following reasons: pure CNFIG matrix exhibits a broad diffraction peak at about 25.9° in the XRD spectrum. This broad peak is very weak because the CNFIG is literally amorphous. In strong contrast, the MoS₂ on the CNFIG is highly crystalline after the sintering treatment at 700°C for 2h, delivering strong XRD response in the XRD spectrum of MoS₂@CNFIG (as seen in Figure S19). Similar phenomena were also observed in other reports, such as Refs. 26, 32 and 34.

Once again, we are grateful for the constructive comments from the reviewers and editor for improving the quality of the manuscript. We wish our effort and the improvement of our work will eventually lead to the publishing of this work in your prestigious journal.

REVIEWERS' COMMENTS:

Reviewer #1 (Remarks to the Author):

The issues raised have been well addressed. This manuscript is acceptable for publication.

Reviewer #2 (Remarks to the Author):

Accept.